# Extracellular S100β Disrupts Bergman Glia Morphology and Synaptic Transmission in Cerebellar Purkinje Cells

**DOI:** 10.3390/brainsci9040080

**Published:** 2019-04-12

**Authors:** Olga S. Belozor, Dariya A. Yakovleva, Ilya V. Potapenko, Andrey N. Shuvaev, Marina V. Smolnikova, Alex Vasilev, Elena A. Pozhilenkova, Anton N. Shuvaev

**Affiliations:** 1Krasnoyarsk State Medical University named after Prof. V.F. Voino-Yasenetsky, Department of Biological Chemistry, Medical Pharmaceutical and Toxicological Chemistry, Partizan Zheleznyak st. 1, 660022 Krasnoyarsk, Russia; olsbelor@gmail.com (O.S.B.); elena.a.pozhilenkova@gmail.com (E.A.P.); 2Siberian Federal University, Svobodny pr., 79, 660041 Krasnoyarsk, Russia; dariayakowlewa@yandex.ru (D.A.Y.); andrey.n.shuvaev@gmail.com (And.N.S.); 3Krasnoyarsk State Medical University named after Prof. V.F. Voino-Yasenetsky, Research Institute of Molecular Medicine and Pathobiochemistry, Partizan Zheleznyak st. 1, 660022 Krasnoyarsk, Russia; iluminator@snkip.ru (I.V.P.); smarinv@yandex.ru (M.V.S.); 4Federal Research Center “Krasnoyarsk Science Center” of the Siberian Branch of the Russian Academy of Sciences, Scientific Research Institute of Medical Problems of the North, Partizan Zheleznyak st., 3G, 660022 Krasnoyarsk, Russia; 5Institute of Living Systems, Immanuel Kant Baltic Federal University, Universitetskaya st., 2, 236041 Kaliningrad, Russia; otherlife@bk.ru

**Keywords:** astrocytes, S100β, Purkinje cells, short-term plasticity, Ca^2+^ signaling

## Abstract

Astrogliosis is a pathological process that affects the density, morphology, and function of astrocytes. It is a common feature of brain trauma, autoimmune diseases, and neurodegeneration including spinocerebellar ataxia type 1 (SCA1), a poorly understood neurodegenerative disease. S100β is a Ca^2+^ binding protein. In SCA1, excessive excretion of S100β by reactive astrocytes and its uptake by Purkinje cells has been demonstrated previously. Under pathological conditions, excessive extracellular concentration of S100β stimulates the production of proinflammatory cytokines and induces apoptosis. We modeled astrogliosis by S100β injections into cerebellar cortex in mice. Injections of S100β led to significant changes in Bergmann glia (BG) cortical organization and affected their processes. S100β also changed morphology of the Purkinje cells (PCs), causing a significant reduction in the dendritic length. Moreover, the short-term synaptic plasticity and depolarization-induced suppression of synaptic transmission were disrupted after S100β injections. We speculate that these effects are the result of Ca^2+^-chelating properties of S100β protein. In summary, exogenous S100β induced astrogliosis in cerebellum could lead to neuronal dysfunction, which resembles a natural neurodegenerative process. We suggest that astrocytes play an essential role in SCA1 pathology, and that astrocytic S100β is an important contributor to this process.

## 1. Introduction

Spinocerebellar ataxia type 1 (SCA1) is a progressive neurodegenerative hereditary disorder that mainly affects the cerebellum and brainstem. It is caused by a dynamic expansion of CAG repeats in the N-terminal coding region of *gene*ATXN1 gene on chromosome 6p23 [1,2]. In the general population, prevalence of SCA1 is only 1–2 per 100,000, but in specific ethnic groups, this ratio varies [3]. SCA1 is characterized by progressive cerebellar dysfunction, dysarthria, and worsening of bulbar functions. Pathological changes include neuronal loss in the cerebellum, brainstem, and degeneration of spinocerebellar tracts [4,5]. Higher cortical functions may also be affected with symptoms including memory loss and verbal and nonverbal intellectual deficits [6]. CAG repeats encode amino acid glutamine; therefore, their expansion leads to the synthesis of Ataxin 1 with excessive polyglutamine tract. This affects protein folding and leads to precipitation as intracellular aggregates [7,8]. The aggregates also contain inclusions of ubiquitin, proteasome components, and chaperons [9]. In the SCA1 mouse model, aggregation occurs later than the first pathological sights appear and does not correlate with the disease severity [10]. Accumulation of the mutant protein leads to selective neurodegeneration in certain regions of the brain and reactive astrocytosis. Astrogliosis in SCA1 is tightly correlated with the onset and severity of disease and is not a consequence of neuronal death. Cvetanovic et al. (2015) described the astrocytic and microglial reaction in SCA1, using non-cell selective SCA1 knock-in (Sca1^154Q/2Q^) and Purkinje cell (PC) selective B05 SCA1^82Q/2Q^ mouse models. They showed that astrocytes and microglia are activated at early stages of SCA1. Expression of protein Ataxin 1 in microglia and astrocytes was not essential for the activation of glia, but expression in Purkinje cells was sufficient for this purpose [11,12].

In cerebellum, reactive astrogliosis of BG may disrupt the spatial distribution of excitatory amino acids transporter 1 (EAAT1) or glutamate aspartate transporter (GLAST). This could result in an increase in extracellular glutamate concentrations and toxicity via N-methyl-D-aspartate (NMDA)-receptors [12,13]. In addition, activated astrocytes and microglia are able to release various proinflammatory molecules. Some of them, such as tumor necrosis factor α (TNFα), interleukin-6 (IL-6), and monocyte chemoattractant protein-1 (MCP-1) have been implicated in neurodegeneration and negatively affect the function and survival of neurons [14,15,16,17,18,19]. S100β protein is one of the S100 Ca^2+^-binding proteins of the S100 group, which includes nearly 20 members [20]. S100 proteins form homo- and heterodimers and are able to chelate not only Ca^2+^, but also Zn^2+^ and Cu^2+^ [21]. Binding of the ions changes the confirmation of S100 and alters their affinity to different ligands (more than 90 potential targets currently known) [10,20,21,22].

Here, we model astrogliosis in cerebellum, such as seen in SCA1, by the intracortical injections of S100β. We demonstrate a significant negative impact of this astrocytic protein to the PC’s morphology and synaptic transmission in the parallel fibre–Purkinje cell (PF–PC) synapse. Moreover, we demonstrate that short-term synaptic plasticity, the depolarization-induced suppression of excitation (DSE), is disrupted by S100β. We speculate that these effects could be attributed to the Ca^2+^-chelating properties of S100β.

## 2. Materials and Methods

All procedures for the care and treatment of animals were carried out according to the Krasnoyarsk State Medical University and Russian public standard (33215–2014) regulations and were approved by the local ethical committee. Every effort was made to minimize animal suffering and to reduce the number of animals used in this study. Twelve-week-old CD-1 IGS WT mice (Charles River Laboratories) were used in this study. Four-week-old non-cell selective SCA1^154Q/2Q^knock-inn (SCA1 KI) mice with C57BL/6J background were used in this work [10]. Experiments with SCA1 KI mice were generated in Gunma University (Japan) in the laboratory of Neurophysiology and Neuronal Repairled by Professor H. Hirai. These were mice kindly provided by Dr. Hidehiro Mizusawa (Tokyo Medical and Dental University). Animals were kept on a 12 h light/dark cycle with free access to food and water.

### 2.1. Drugs and Reagents

All reagents for electrophysiological experiments were from Sigma Aldrich.

Recombinant mouse S100 Calcium Binding Protein B (S100β) (Cat. No. APA567Mu01) was obtained from Cloud-Clone Corp. A concentrated stock solution of S100β was initially prepared and diluted in physiological saline (PBS) to a final concentration before use. Fluorocitrate (FC) was used as barium salt (Cat. F9634, Sigma Aldrich) as described previously (Paulsen et al., 1987). (3S)-3-[[3-[[4-(Trifluoromethyl)benzoyl]amino]phenyl]methoxy]-L-aspartic acid (TFB-TBOA) (Cat. No. 2532) was obtained from Tocris. Concentrated stock solutions of FC and TBOA were initially prepared and diluted in artificial spinocerebellar fluid (ACSF) to their final concentrations before use.

### 2.2. S100β Injections

Twelve-week-old (P90-P100) WT CD1 mice were anaesthetized by intraperitoneal injection of chloral hydrate solution (400 mg/kg of body weight). Then, 2.5 μL of 50 μM S100β in phosphate buffered saline (PBS) or PBS was stereotaxically injected into the cortex of cerebellar vermis (lobule VI) using a 10 μL Hammilton syringe. To reach the injection point in the vermis, we used the coordinates relative to bregma: AP: −2.5 mm, ML: 0 mm, DV: 2 mm. Mice were used 24 h after the injection.

### 2.3. Electrophysiology

Cerebellar slices (250 μm thick) were prepared, and whole-cell recordings were conducted as described previously [23]. Briefly, mice were deeply anesthetized by intraperitoneal injection of chloral hydrate (400 mg/kg of body weight) and killed by decapitation. The brain was quickly dissected and placed for one minute in an ice-cold Ringer’s solution containing the following: 234 mM sucrose, 26 mM NaHCO_3_, 2.5 mM KCl, 1.25 mM NaH_2_PO_4_, 11 mM glucose, 10 mM MgSO_4_, and 0.5 mM CaCl_2_ 0.5; pH 7.4, continuously oxygenated with 95% O_2_ and 5% CO_2_. Parasagittal slices of cerebellar vermis were made using a microslicer (Thermo Scientific; Microtom CU65). The slices were maintained in an extracellular solution containing the following: 125 mM NaCl, 2.5 mM KCl, 2 mM CaCl_2_, 1 mM MgCl_2_, 1.25 mM NaH_2_PO_4_, 26 mM NaHCO_3_, 10 mM D-glucose, and 0.05–0.1 mM picrotoxin bubbled by 95% O_2_/5% CO_2_ gas mix at room temperature for 1h before starting the electrophysiological experiments. For current clamp whole-cell recordings from Purkinje cells (PCs), we used K-gluconate-based intracellular solution containing the following: 130 mM K-gluconate, 4 mM KCl, 20 mM HEPES, 1 mM MgCl_2_, 4 mM MgATP, 1 mM NaGTP, and 0.4 mM EGTA (pH 7.3 adjusted with KOH). For voltage clamp whole-cell recordings from Purkinje cells (PCs), we used intracellular solution containing the following: 140 mM Cs-gluconate, 8 mM KCl, 10 mM HEPES, 1 mM MgCl_2_, 2 mM MgATP, 0.4 mM NaGTP, and 0.4 mM EGTA (pH 7.3 adjusted with CsOH). Passive electrical properties of the PCs were estimated using averaged traces of ~10 current responses to hyperpolarising voltage pulses (from −70 to −80 mV, 200 ms duration). Fast capacitance component was automatically compensated; the signal was sampled at 50 kHz and low-pass filtered at 10 kHz. No correction was made for liquid junction potentials. Analysis of electrophysiological data was performed using pClamp10 (Molecular Devices), Pachmaster software (HEKA), and Clampfit 10.5 (Axon instruments).

PCs were voltage-clamped at −70 mV to record excitatory postsynaptic currents after irritation of parallel fibers (PF EPSCs). Selective stimulation of PFs was confirmed by paired-pulse facilitation of EPSC amplitudes (at a 50 ms interstimulus interval).

To examine depolarization induced suppression of excitation (DSE), PF EPSCs were recorded every 3 s. After monitoring basal PF EPSCs for 1 min, a single depolarizing pulse (5s from −70 to 0 mV) was applied to the recorded PC. This opens the voltage gated Ca^2+^ channels (VGCC) and releases endocannabinoids, which presynaptically decrease glutamate release and suppress amplitude of PF EPSC [24]. Amplitudes of subsequent PF EPSCs were normalized to the mean value of 12 responses evoked before the induction of DSE.

### 2.4. Immunohistochemistry

For immunohistochemistry (IHC), anesthetized mice were perfused transcardially with 4% paraformaldehyde in 0.1 M phosphate buffer. The brain was postfixed in the same fixative overnight. The cerebellar vermis was cut into 50 µm sagittal sections. The sections were treated with rabbit monoclonal anti-calbindin D-28 k (1:500, Cloud Clone Corp., China), chicken polyclonal anti-GFAP antibodies (1:1000, Abcam, UK), and rabbit polyclonal anti-S100β (1:1000, Abcam, UK). Secondary antibodies were Alexa Fluor 514-conjugated donkey anti-rabbit IgG (1:1,000, Life Technologies), Alexa Fluor 647-conjugated donkey anti-chicken IgG (1:1,000, Life Technologies), and Alexa Fluor 488-conjugated donkey anti-rabbit IgG (1:1,000, Life Technologies). Antibodies were dissolved in PBS solution containing 2% (*v*/*v*) normal donkey serum, 0.1% (*v*/*v*) Triton X-100, and 0.05% NaN_3_.

### 2.5. Confocal Microscopy and Morphometric Analysis

Fluorescent images were obtained using FV10i Confocal Laser Scanning Microscope and the original software “Fluoview” (Olympus, Japan). Images were recorded as Z-stacks of 0.25 μm thickness with a ×10 lense, numerical aperture of 1.0, zoom ×6, and 1024 × 1024 resolution. In all groups, the cerebellar lobes 6 and 7 of the vermis cerebellum were used for comparison (Figure 1). For double labeling, images from the same confocal plane were taken. Alexa Fluor 647 signal (blue) was artificially changed to red color to show S100β/GFAP colocolization in merge microphotographs (yellow) (Figure 1 and Figure 2).

The thickness and number of BG processes were measured on confocal images of sagittal cerebellar slices. The number of radial glial processes for 100 µm of the molecular layer (Appendix A) was counted. The same 100 µm line intensity profile was used to obtain the distribution of GFAP fluorescence, using the original software of the Olympus confocal microscope. Each glial process was shown as a peak of GFAP/Alexa 647 fluorescence intensity. We counted the averaged thickness of these processes in each image. To avoid false positive enhancement of the process’s thickness, we used cut-off threshold for recognition of the GFAP signal set to 10% of the maximal fluorescence intensity. To count the number of BG cells, we measured anti-S100β positive circle- and oval-shaped signals in the Purkinje cell layer. To avoid over- or underestimation, the glial cell numbers were calculated by ceiling the ratio $n=\lceil \frac{d}{d^{\prime }}\rceil$, where n is the cell number; d is the length of S100β positive signal in μm; and $d^{\prime }$ = 15 μm, which we took for characteristic diameter of an astrocyte. The approximate length of the dendrites of the Purkinje cells was estimated from overall thickness of the molecular layer (Appendix A), visualized using anti-calbindin/Alexa 488 staining.

### 2.6. Sholl Analysis of Bergmann Glia Cells

Quantitative morphological analysis was performed in the three-dimensional (3D) mode. Using a confocal laser scanning microscope (Olympus, Fluoview, FV10i), anti-GFAP-labeled BG was scanned in Z-stacks (80–150 consecutive focal planes at 0.25 μm intervals). For Scholl analysis, Z-stacks mages of soma and processes of BG were traced on focal planes using ImageJ software. We used the Sholl method of concentric circles using an ImageJ regime (set of nested concentric spheres is centered on the cell body, and the spheres increase in size by 10 μm radius) [25]. The results of the Sholl analysis showed the length of processes and the number of intersections per 10 μm.

### 2.7. Statistical and Mathematical Analysis

Pooled data are expressed as the mean ± standard error of mean (SEM). Statistical analyses of differences between the groups were performed using the unpaired t-test and Mann–Whitney U test. The influence of FC and TBOA on the EPSC recovery was estimated with a one-way analysis of varaiance (ANOVA) test. Differences were considered significant at *p* < 0.05.

We estimated the dendritic and somatic capacitance by optimization of the two-term exponential series to the current response curve to the voltage step to find the time constants $\tau_{i}$. Here, $R_{ss}=4$ MΩ is the input resistance and $A_{i}$ are the free parameters. Indices d and s stand for dendritic and somatic components, respectively. The resulting capacitance was then calculated as Ci=τiRm{i=d,s}. $R_{m}$ is the membrane resistance. Optimization was made in ClampFit 10.7 software.

$V_{step}=10$ mV [26]:

Iclamp(t)=Vstep(1Rss−Adτde−t/τd−Asτse−t/τs).

DSE was analyzed using dual exponential waveform Equation (1).(1)$$DSE=100+A(e^{\frac{-t}{\tau_{1}}}-e^{\frac{-t}{\tau_{2}}})\{A=\frac{100a\tau_{1}\tau_{2}}{\tau_{1}-\tau_{2}}\}$$

This curve is convenient for the prediction of the conduction changes in synapses [27]. It contains the parameters for both decay and recovery of EPSC separately during the DSE protocol. This model was fitted to the experimental data by the Nelder–Mead minimization of the sum of squared residuals to find “A”—the maximum EPSC decrease in percent of initial level, and “*τ*_1_” and “*τ*_2_”—the half-times for the EPSC to reach the minimum and to recover to the initial 100%, respectively. The parametric bootstrap technique was used to obtain the 95% confidence intervals for the parameters *A*, *τ*_1_, and *τ*_2_. This analysis was performed using the Python 3 package.

## 3. Results

### 3.1. Exogenous S100β Alters Morphology of Bergmann Glia

First, 2.5 µL of 50 µM S100β was injected in the cerebellar cortex to induce astrogliosis. Twenty-four hours later, widespread distribution of S100β and increased GFAP expression in cerebellar cortex were evident in lobules IV–VII (Figure 1C compared with Figure 1A; note that in the normal brain, immunofluorescent GFAP was poorly visible). In microphotographs of S100β injected areas, prominent GFAP positive striation was evident (contrast to PBS injected areas, Figure 1B,D). To analyze the morphology of BG, we examined the number and thickness of anti-GFAP-positive glial processes in the central part of molecular layer per 100 μm using the line profile function (Appendix A and Figure 2A,B). The average cross-section of BG processes in S100β injected mice was increased to 3.6 ± 0.1 µm (364 processes from 17 areas of 3 mice) versus 2.8 ± 0.1 µm (358 processes from 11 areas of 3 mice) in PBS injected mice, *p* = 1.25 × 10^−13^, unpaired *t*-test (Figure 2C). The number of processes per 100 μm longitudinal length of molecular layer in S100β injected animals was significantly decreased compared with PBS injected animals (21.4 ± 2.0 vs. 32.6 ± 3.3, *p* = 0.013, *t*-test; Figure 2D). The density of BG processes was also decreased in S100β injected areas. We also measured the fraction of “GFAP-negative” space in the central part of molecular layer per 100 μm using the same line profile function. In S100β injected areas (17 areas of 3 mice), it was increased to 24.0% ± 4.5% compared with 8.1% ± 1.6% in PBS-injected areas (11 areas of 3 mice, *p* = 0.015; unpaired *t*-test; Figure 2E). The low BG process’s density is mainly the result of the loss of some of these cells. Indeed, the number of anti-S100β-labeled cell bodies per 100 μm longitudinal length of Purkinje cell layer in S100β injected animals was significantly decreased in comparison with PBS injected animals (9.1 ± 0.4 vs. 10.9 ± 0.5, *p* = 0.007, unpaired *t*-test; Figure 2F).

Next, we studied single astrocyte morphology using Sholl analysis (Sholl, 1953). On images of digitally traced BG processes (Figure 3A), we analyzed the maximum number of these processes per cell. This number was not changed in S100β injected areas (4.9 ± 1.2, *n* = 12 from 3 animals) in comparison with PBS-injected areas (4.8 ± 1.7, *n* = 10 from 3 animals, *p* = 0.648; Mann–Whitney U test; Figure 3B). Sholl analysis revealed an increase in the density of proximal processes in BG after S100β injections. Within 10 μm from soma in S100β injected areas (12 areas of 3 mice), BG had 3.3 ± 0.3 processes, while in PBS-injected areas, it was 1.8 ± 0.2 (*p* = 0.0002; 11 areas of 3 mice, Mann–Whitney U test; Figure 3C).

These data show that excessive extracellular S100β protein in the cerebellar cortex leads to significant changes in BG morphology.

### 3.2. Extracellular S100β Alters Morphology of Purkinje Cells

As shown above, S100β affects glia and it is well-known that disturbances in glia may lead to neuronal degeneration [28,29,30]. In addition, S100β could have a direct effect on Purkinje cells. We examined the effect of S100β on morphology of these neurons using IHC and their physiological state using patch clamp. To estimate the approximate dendritic length of PCs, cells were visualized by anti-calbindin staining (Figure 4A) and the thickness of the molecular layer was measured (Appendix AB). S100β injections reduced it to 120.0 ± 5.8 µm (*n* = 12 areas from 3 mice) compared with 150.7 ± 6.3 µm, *n* = 14 areas from 3 mice in PBS-injected mice (*p* = 0.002, *t*-test; Figure 4B).

Using patch clamp, we estimated capacitance of dendrites and soma after subtraction of slow capacitance component from the total capacitance of PCs (see materials and methods). A slow component predominantly reflects the size of neuronal dendrites. We found a significant difference between the two groups. The capacitances of PCs’ dendrites in S100β injected mice were 359.4 ± 37.5 pF (*n* = 33 cells from 8 mice) and 513.5 ± 27.1 pF (*n* = 52 cells from 10 mice) in the PBS-injected group. (*p* = 0.002, *t*-test; Figure 3B). The capacitances of PCs’ soma in S100β injected mice were 34.6 ± 4.4 pF (the same cells) and 61.7 ± 5.6 pF (the same cells) in the PBS-injected group. (*p* = 0.0003, *t*-test; Figure 3C).

These data indicate that excessive extracellular S100β affects PCs’ morphology, leading to the collapse of the soma and dendrites.

### 3.3. Extracellular S100β Alters Synaptic Transmission in PFs and PCs

Astrocytes control the removal of glutamate from the presynaptic space [31,32,33,34]. Moreover, astrocytic secretion of S100β protein into the intercellular space leads to endocytosis of this protein by neurons and evokes various effects such as chelation of cytoplasmic Ca^2+^ [20]. For this reason, we tested whether elevated extracellular S100β affects synaptic transmission in PF–PCs synapses. S100β did not change the PF-EPSC’s amplitude (Appendix A). We suspected that S100β will affect processes highly dependent on Ca^2+^ release, such as presynaptic glutamate secretion. However, we did not see significant differences between the paired pulse facilitation (PPF) ratio in PF–PC synapses of S100β and PBS-injected mice. The PPF ratio in S100β injected mice was 1.8 ± 0.4, *n* = 35 cells from 8 mice versus 1.9 ± 0.1, *n* = 39 cells from 9 mice in PBS-injected mice (*p* = 0.722, *t*-test; Figure 5A). However, S100β dramatically affected the kinetics of PF-EPSCs. While there was no statistically significant difference in the PF-EPSC’s amplitude in mice injected with S100β and PBS (Appendix A), the rise time of PF-EPSC in S100β injected mice was prolonged to 2.7 ± 0.1 ms (*n* = 35 cells from 8 mice), compared with 2.3 ± 0.1 ms (*n* = 36 cells from 9 mice) in PBS-injected mice (*p* = 0.028, *t*-test; Figure 5B).

To control for the potential effect of surgery per se, we analyzed this parameter in sliced from naïve mice. No difference was found between naïve and PBS injected groups (Appendix A).

In addition, the decay time of PF-EPSC in S100β injected mice was 17.1 ± 1.5 ms (*n* = 30 cells from 8 mice), while it increased to 21.6 ± 1.5 ms (*n* = 37 cells from 9 mice) in PBS-injected mice (*p* = 0.04, *t*-test; Figure 5C).

These results demonstrate that the excessive extracellular accumulation S100β protein mainly affects kinetics of PF-EPSC, which most likely reflects changes glutamate removal from the synaptic cleft.

### 3.4. Similarities in Changes in Synaptic Transmission in S100β-Injected Mice and Ataxin1 Mutant Animals

To look for similarities between SCA1 and consequences of S100β injections, we used KI mice with non-cell selective expression of mutant Ataxin 1 [10]. Mice were used at three weeks of age, which corresponds to the early stage of the neurodegenerative process. There was no statistically significant difference in the PF-EPSCs amplitude recorded in the PCs of SCA1 KI and WT mice (Appendix A). The PPF ratio in SCA1 KI mice was 2.1 ± 0.1, *n* = 10 cells from 3 mice and 1.8 ± 0.1, *n* = 8 cells from 3 mice in WT mice (*p* = 0.012, unpaired t-test; Figure 6A). PCs from SCA1 KI mice have altered kinetics of PF-EPSCs. The average rise time of PF-EPSC in SCA1 KI mice significantly increased to 3.0 ± 0.2 ms (*n* = 10 cells from 3 mice), compared with WT mice (2.3 ± 0.2 ms; *n* = 8 cells from 3 mice; *p* = 0.038, unpaired *t*-test; Figure 6B). The differences in decay time of PF-EPSC between SCA1 KI mice and WT were not significant (18.4 ± 2.1 ms, *n* = 10 cells from 3 mice vs. 15.9 ± 3.3 ms, *n* = 8 cells from 3 mice; *p* = 0.379, unpaired *t*-test; Figure 6C).

### 3.5. Extracellular S100β Alters Endocannabinoid-Dependent Short Term Plasticity in PF–PC Synapses

As mentioned above, we expected that S100β could affect processes that are known to depend on the cytosolic concentration of Ca^2+^. One such process is DSE, which is evoked by the membrane depolarization. Depolarization leads to the opening of voltage-gated calcium channels (VGCC) and an increase in the intracellular Ca^2+^. Ca^2+^ triggers endocannabinoid release from the postsynaptic cell with consecutive activation of CB1 receptors on the presynaptic terminal, leading to a reduction in release of glutamate. We examined dynamic of PF-EPSC amplitude after 5 s of depolarization from −70 to 0 mV. The stimulus intensity was adjusted to reach an EPSC amplitude of approximately 150 pA before DSE induction. In control mice, DSE protocol reduced EPSC by 67.3% ± 3.5% (*n* = 11 cells from 4 mice), which was similar to that in mice pre-injected with S100β (69.7% ± 4.7%, *n* = 12 cells from 4 mice, *p* = 0,975, unpaired *t*-test; Figure 7A). However, the recovery of the PCs’ amplitude was significantly faster in S100β injected mice. Fifty seconds after the challenge, amplitude returned to 93.6% ± 2.8% of control, compared with 83.9% ± 2.7% in PBS-injected mice (*p* = 0.03, unpaired *t*-test; Figure 7A,B).

The double waveform model fit (1) confirmed the slowing of the recovery kinetics by S100β. The amplitude reduction in the DSE protocol was not significantly different in S100β-injected mice compared with the PBS group: 45.5% (30.6%, 63.6%) versus 38.1% (31.8%, 44.6%), respectively. Using the fitting protocol, we calculate that the half-time for the recovery of the parameter (*τ*_2_) is significantly smaller in S100β-injected mice with 29.7 s (21.0 s, 47.2 s) in comparison with 64.5 s (52.0 s, 85.1 s) in the PBS group, *p* < 0.05.

These results demonstrate that excessive extracellular S100β protein negatively affects DSE.

### 3.6. Effects of FC on PF–PC Transmission and Endocannabinoid Short Term Plasticity in PF–PC Synapses

FC inhibits astrocytic metabolism and deprives these cells of energy, leading to an array of repercussions, which ultimately undermine the functions of these cells [35]. The application of FC led to a strong depression of PF–PC excitatory transmission, irrespective of whether the tissue was exposed to S100β or not (Figure 8A,B). After 10 min application, the amplitude of PF-EPSC in S100β injected mice decreased to 57.7% ± 9.0% of the control (*n* = 7 cells from 4 mice, *p* = 0.006, paired *t*-test) and in PBS-injected mice, to 64.4% ± 9.9% of the control (*n* = 7 cells from 4 mice, *p* = 0.014, paired *t*-test) (Figure 8A,B). Ten minutes after FC application, averaged PF-EPSCs in S100β- and PBS-injected groups were not different (unpaired *t*-test, *p* = 0.65). The rise time of PF-EPSC in PBS- and S100β-injected mice increased after FC treatment. In the PBS group, it increased from 2.3 ± 0.2 ms to 2.7 ± 0.2 ms (*n* = 12 cells from 4 mice, *p* = 0.003, paired *t*-test; Figure 8C), while in the S100β group, it increased from 2.4 ± 0.3 ms to 3.5 ± 0.6 ms (*n* = 10 cells from 3 mice, *p* = 0.003, paired *t*-test; Figure 8C).

It was shown previously that astrocytes also contain CB1 receptors and could modulate the synaptic plasticity [36]. CB1 receptors in astrocytes are coupled to G_q/11_-proteins and trigger PLC activation [37] and release such gliotransmitters as glutamate, ATP, or d-serine [38]. This phenomenon is SNARE-dependent and highly sensitive to ATP concentration, which should be reduced by FC [39,40]. However, FC did not affect the expression of DSE. After FC application in PBS-injected mice, DSE protocol reduced EPSC by 56.8% ± 8.7% (*n* = 9 cells from 4 mice), which was not statistically different from that registered without FC in S100β-injected mice (Figure 8E) (*p* = 0.317, unpaired *t*-test). However, this comparison is compromised by the direct impact of FC on PF-induced EPSCs and should be interpreted with care.

The double waveform model fits the changes of PF-EPSC amplitudes after depolarization pulse and DSE initiation. Maximum amplitudes reduction was 51.92% (41.67%, 81.41%) and 46.23% (33.28%, 75.49%) for PBS and S100β, respectively.

However, the recovery of the PCs’ amplitude was significantly slower in FC-treated PCs in slices from PBS-treated mice (Figure 8F). Fifty seconds after the challenge, it returned to 67.5% ± 5.4% of control, compared with untreated slices, where it recovered to 83.9% ± 2.7%, *p* = 0.025, unpaired *t*-test; Figure 8D–F. However, FC had hardly any effect on DSE protocol in S100β injected mice, where recovery of the PCs’ amplitude was 81.4% ± 7.6% of the PF-EPSC amplitude at 50 s post challenge in FC versus 93.6% ± 2.8% without FC; *p* = 0.181, unpaired *t*-test (Figure 8D–F).

The double waveform model fit also did not reveal significant changes after the FC treatment.

### 3.7. Slowdown of Glutamate Uptake in Bergmann Glia by TBOA Leads to Alteration of PF-EPSC Kinetic, But Does Not Change Endocannabinoid Short Term Plasticity in PF-PC Synapses

Ninety percent of all glutamate uptakes in PF–PC synapses is the result of excitatory amino acid transporters EAAT1 and EAAT2, located on the membranes of astrocytes [41,42]. We suspected that shortening of PF-EPSC decay time after S100β injection was the result of the facilitation of glutamate reuptake through EAATs (Figure 5C). We found that 500 μM DL-threo-β-Benzyloxyaspartic acid (TBOA) significantly increased the decay time of PF-EPSC in PBS- and S100β-injected mice. In PBS-injected mice, the decay time changed from 30.0 ± 4.4 to 39.2 ± 6.3 msec (*n* = 16 cells from 4 mice, *p* = 0.039 paired *t*-test, Figure 9A). Note that before TBOA application, decay constants were different between S100β- and PBS-injected groups (*p* = 0.044, one-way ANOVA), while after TBOA, they reached approximately the same values (*p* = 0.64, one-way ANOVA, Figure 9A). In S100β-injected mice, the increase was even more dramatic, from 19.2 ± 2.7 ms to 34.6 ± 7.1 ms (*n* = 10 cells from 3 mice, *p* = 0.031 paired *t*-test). Application of nonselective EAAT blocker DL-TBOA 500 μM did not significantly change PF-EPSC amplitudes and rise time in the cerebellum of both PBS- and S100β-injected mice (data not shown). TBOA did not affect the PPF ratio of PF-EPSC in PBS-injected mice (1.86 ± 0.1 vs. 1.84 ± 0.1; *n* = 16 cells from 4 mice, *p* = 0.673 paired *t*-test), while in S100β-injected mice, it resulted in a slight, but significant PPF ratio reduction from 1.9 ± 0.2 to 1.76 ± 0.1 (*n* = 10 cells from 3 mice, *p* = 0.049 paired *t*-test, Figure 9B).

We examined endocannabinoid short-term plasticity in PF–PC synapses after treatment of cerebellar slices with 500 μM TBOA. In PBS-injected mice, DSE protocol was not affected by TBOA. EPSC was reduced by 59.2% ± 5.4%; *n* = 15 cells from 4 mice), which was not statistically different from that before application (56.8% ± 5.7%, *p* = 0.524, paired *t*-test). Furthermore, the recovery of the PCs amplitude was not affected by TBOA (Figure 9C,D), nor did TBOA affect the outcome of DSE protocol in S100β-injected mice (74.8% ± 9.6% reduction in TBOA, *n* = 11 cells from 4 mice vs. 73.6% ± 8.8% without TBOA, *p* = 0.789, paired *t*-test). Also, the recovery of the PCs amplitude was unaffected by TBOA. The double waveform model yielded in the PF-EPSC amplitudes reduction during DSE as 45.71% (37.39%, 56.75%) and 32.11% (14.73%, 75.00%) for PBS and S100β cells, respectively. The restoration half-time $\tau_{2}$ was 69.28 s (53.08 s, 92.12 s) for PBS cells and 46.72 s (18.68 s, 113.44 s) for S100β cells. These results confirm that TBOA did not affect the outcome of the DSE (Figure 9C,D).

These results demonstrate that the slowdown of glutamate uptake predictably affects PF-EPSC decay time and does not change DSE [10].

## 4. Discussion

It was shown previously that activated astrocytes secrete large amounts of S100β [11,18]. PCs absorb glial S100β in cytoplasmic vacuoles, which leads to changes in their morphology and degeneration [19,43]. In the SCA1 B05 transgenic (tg) mouse model, the formation of S100β-containing cytoplasmic vacuoles precedes the accumulation of the mutant Ataxin 1 and appearance of the ataxic phenotype [11]. Downregulation of S100β rescues the neurological deficit; therefore, it has been argued that this protein plays the central role in neurodegeneration [44]. We attempted to mimic some of the features of SCA1 by injections of S100β. Mouse S100β was used in order to avoid an immune reaction to a foreign antigen. Using immunohistochemistry, we found that the area of S100β deposits spread much further than the actual site of injection and covered 2–3 lobes of the cerebellum. The maximal immunochistochemical signal was seen in lobes 5–7 (Figure 1C). For imaging, we adjusted the brightness of S100β/Alexa 488 fluorescence signal so as to be able to image high concentration S100β without saturation of the system. For this reason, the fluorescence of the endogenous S100β in the control mice appears low (Figure 1A,B). To avoid false positive results when measuring the BG and PC morphology, we used lobes 6 and 7, which were not directly affected by the injection in all experiments. The excessive amount of S100β altered the morphology of BG. Reduction of BG cell number (Figure 2B,F) and processes (Figure 2D) was accompanied by thickening of the processes (Figure 2C) and sprouting of new processes in presomatic areas (Figure 3A,C).

The thickness of the molecular layer correlates with the length of the PC dendritic tree [45]. To examine the approximate length of PCs’ dendrites, we measured the thickness of the molecular layer and found that it was significantly reduced by S100β (Figure 4A,B). Accordingly, soma and dendritic capacitance of PCs also changed significantly (Figure 4C).

It is interesting that S100β may modulate sodium channels in neurons via Ca^2+^ chelation, which affects neuronal sodium channels [46], which may result in bursting. Kolta with coauthors showed that Ca^2+^ chelation by S100β in intercellular space leads to Na^+^ current enhancement and makes them fire action potentials in bursts, rather than single AP. It is very likely that this opens NMDA receptors and brings more Ca^2+^ inside the neurons [46]. Such a mechanism could lead to Ca^2+^ overload of neurons, potentially contributing to neurotoxicity in our model.

The dendritic tree of each PC has thousands of synaptic connections with parallel fibers and 1–2 synapses with climbing fibers [47]. Hence, abnormality of dendritic morphology could lead to alteration of basic synaptic transmission, such as EPSC. Our data suggest that S100β did not affect presynaptic glutamate release, because it did not significantly change the PF-EPSC amplitude (Appendix A) and PPF ratio (Figure 5A). Interestingly, in the PCs of three-week-old SCA1 KI mice, where Ataxin-1 is ubiquitously expressed, the PPF ratio was increased in comparison with its WT littermates (Figure 6A). We suspect than this phenomenon is because of the slowing of glutamate–glutamine cycle machinery, because the application of FC leads to the same effect in PBS- and S100β-injected PCs (Figure 8C). We also show that the PF-EPSC rise time is increased in three-week-old SCA1 KI mice (Figure 6B). It is notable that in B05 mice that express mutant Ataxin 1 selectively in PCs, the rise time was not changed at the same age [48]. Therefore, it is likely that changes caused by Ataxin 1 in other cells such as Muller glia are responsible for this effect. Injections of S100β accelerated PF EPSCs’ decay time (Figure 5C), which also points to the involvement of the BG, which plays the key role in glutamate uptake [49,50]. The application of TBOA, which blocks this uptake, ameliorated the differences between PF-EPSC decay time of PCs in PBS and S100β areas, consistent with this hypothesis (Figure 9B).

At the same time, astrogliosis and neurodegeneration is characterized by suppression of EAAT1 and EAAT2 function and accumulation of extracellular glutamate, leading to excitotoxicity [50]. It highly likely that with age, the tendency for prolongation of PF-EPSC decay time SCA1 KI mice (Figure 6C) will increase and become significant. Therefore, we acknowledge that our S100β injection model may not fully reflect the complex pathological process in BG.

The effect of S100β on short-term synaptic plasticity was assessed using DSE. In DSE, strong depolarization leads to Ca^2+^-dependent release of endocannabinoids from PCs, which retrogradely activate the CB1 receptors on the terminals of PFs [51]. Activation of CB1 inhibits glutamate vesicular release. S100β did not alter PPF ratio, which suggests that as such, the vesicular release machinery remained intact (Figure 5A). It is acknowledged that CB may also have direct effects on glia [52], but obviously under our conditions, we did not reveal this component.

Our modelling also confirmed that while DSE protocol depolarization was sufficient to decrease PF-EPSC amplitude up to 67%–69% of control in S100β- and PBS-injected mice (Figure 7A), there was no difference between these two groups. Thus, the induction phase of DSE was not affected by S100, but the recovery after the initial depression was much faster in S100β injected slices. In PBS-injected animals, the amplitude of PF-EPSC was fully restored to the control level at ~100 s after depolarization, but it only took 60–70 s in the S100β-injected group. The predicted speed of restoration by the double waveform fitting (shown by solid and dashed lines on Figure 7A) resulted in the significantly shorter recovery half-time in S100β group compared with the PBS. Endocannabinoids are degraded by fatty acidamide hydrolase and monoacylglycerol lipase. Faster recovery from DSE could be a result of upregulation of these enzymes or could simply indicate that S100β affected the process of endocannabinoid release. A possible explanation for the increased PF-EPSC recovery is an increase in extracellular glutamate concentration. In astrocytes, CB receptors act as antagonists of neuronal CB1 receptors and facilitate the neurotransmitter release [39,52]. Possibly, S100β-activated astrocytes release more glutamate to synaptic cleft after their CB receptors activation by DSE. As such, we used FC to evoke metabolic “starvation” of astrocytes. Interestingly DSE (Figure 8E) in PBS-injected areas was sensitive to FC, but in S100β, it was not. This suggests that the reactive astrocytes after S100β injection do not respond to endocannabinoids. These data correlate with previous findings [52,53].

In summary, elevated extracellular S100β leads to reorganization of glia/neuron morphology and disturbs synaptic transmission. Our findings are reminiscent of the early stage of a neurodegenerative process in cerebellar cortex, such as that seen from three-old-week SCA1 mice [23]. The changes in PF-EPSC kinetics reported here were not seen in the non-cell selective SCA1 model mice, but take place in PC selective SCA1 model mice, where astrocytes are also affected by the mutant Atxn1 ([10] and Figure 6B). We hope that our model will assist in the better understanding of the role of glia in SCA1 and other diseases that affect the cerebellum.

## Figures and Tables

**Figure 1 brainsci-09-00080-f001:**
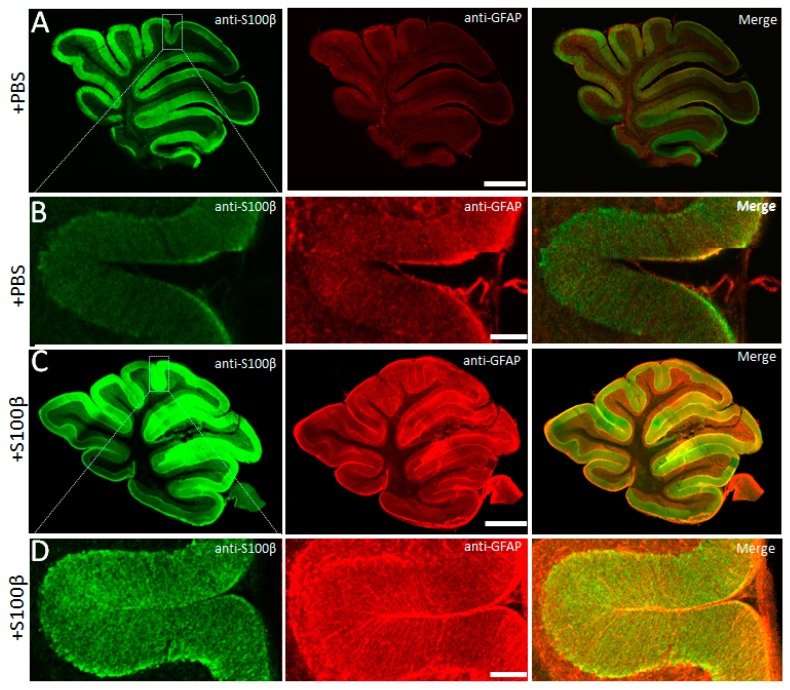
S100β in mouse cerebellum. (**A**) Confocal image of a cerebellar slice injected with phosphate buffered saline (PBS). Left: anti-S100β. Right: anti-GFAP. Scale bar 1 mm. (**B**) High power confocal image of the zone indicated on (**A**). Scale bar 100 μm. (**C**) and (**D**) same as above, but from mice pre-injected with S100β. Scale bar in (**C**) is 1 mm, and in (**D**) is 100 μm.

**Figure 2 brainsci-09-00080-f002:**
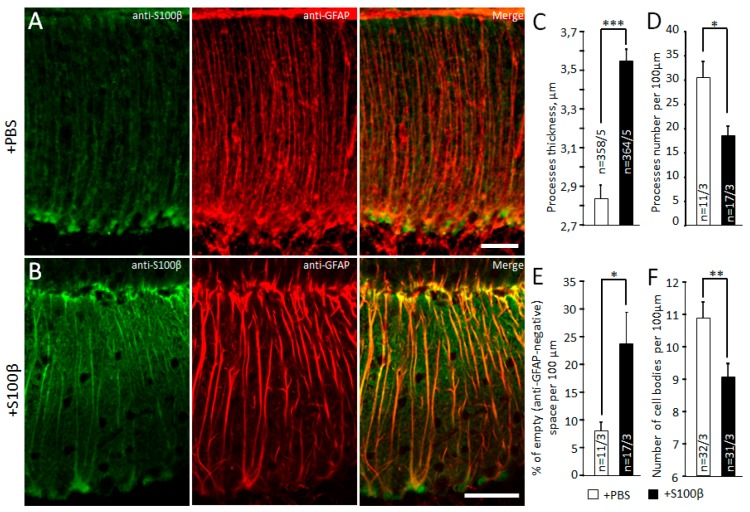
S100β alters morphology of Bergmann glia. Microphotographs show anti-S100β and anti-GFAP immunoreactivity in injected areas from PBS (**A**) and S100β (**B**) treated mice. Scale bar 50 μm. (**C**) Summary graph showing the thickness of BG processes in μm; *** *p*< 0.001. (**D**) In comparison with PBS injected areas, there were significantly fewer BG processes per 100 μ in S100β injected areas; * *p* < 0.05. (**E**) BG processes were sparser in S100β injected areas; * *p* < 0.05. (**F**) Reduction in BG cell bodies caused by S100β; ** *p* < 0.01.

**Figure 3 brainsci-09-00080-f003:**
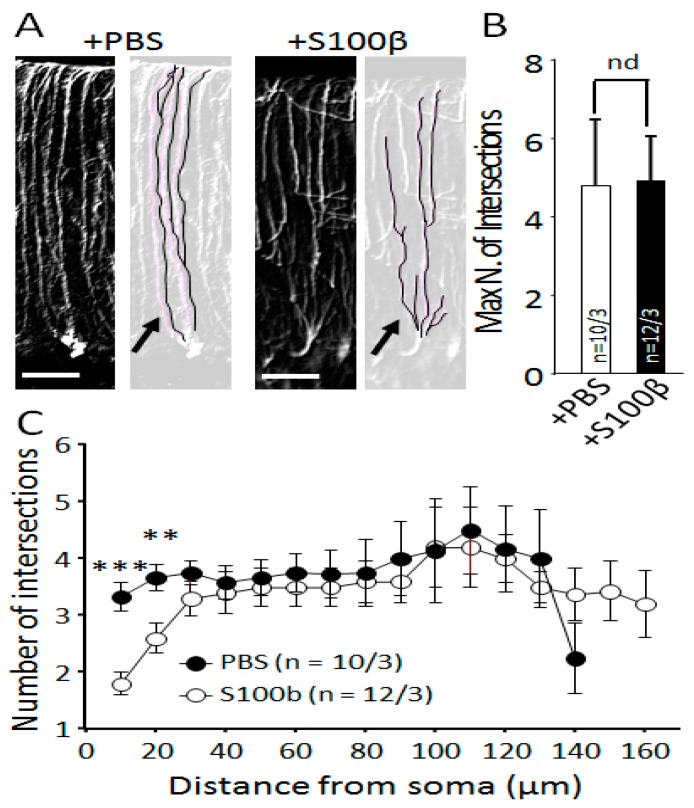
Analysis of Bergmann glia process morphology in PBS and S100β-injected mice. (**A**) White and black microphotographs of anti-GFAP-labeled areas of cerebellar cortex injected with PBS (left images) and S100β (right images). Light images contain digitally traced BG processes generated using ImageJ software. Arrows show the proximal processes that were found more often in S100β injected areas. Scale bar 50 μm. (**B**) Maximum number of processes per BG cell did not change between PBS and S100β-injected areas. (**C**) Sholl analysis for PBS- (open circles) and S100β-injected (closed circles) BG cells (number of intersections per 10 μm of processes length). ** *p* < 0.01, *** *p* < 0.001.

**Figure 4 brainsci-09-00080-f004:**
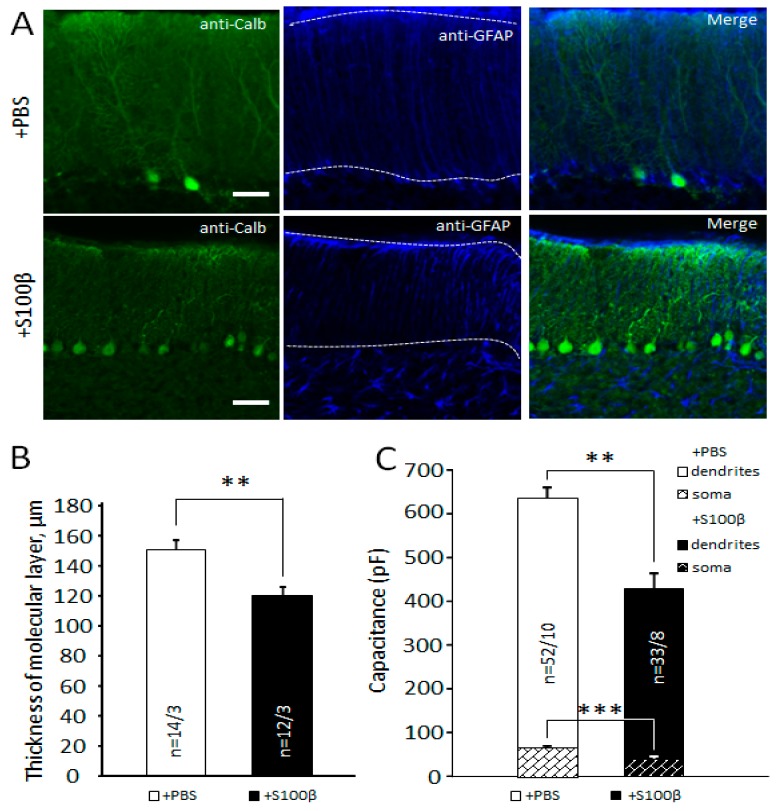
Alters the Purkinje cells’ (PCs’) morphology of mouse cerebellum. (**A**) Comparison of immunoreactivity to anti-calbindin (PCs’ marker), and anti-GFAP in injected areas with PBS (upper panel) and S100β (lower panel). The molecular layer selected by white broken lines. Scale bar 50 μm. (**B**) In comparison with PBS-injected areas, the molecular layer was significantly thinner (79.6%) in S100β injected areas. (**C**) In comparison with PBS-injected areas, the capacitance of PCs’ dendrites and soma measured by voltage-clamp was significantly smaller in S100β injected areas. ** *p* < 0.01, ** *p* < 0.01, *** *p* < 0.001.

**Figure 5 brainsci-09-00080-f005:**
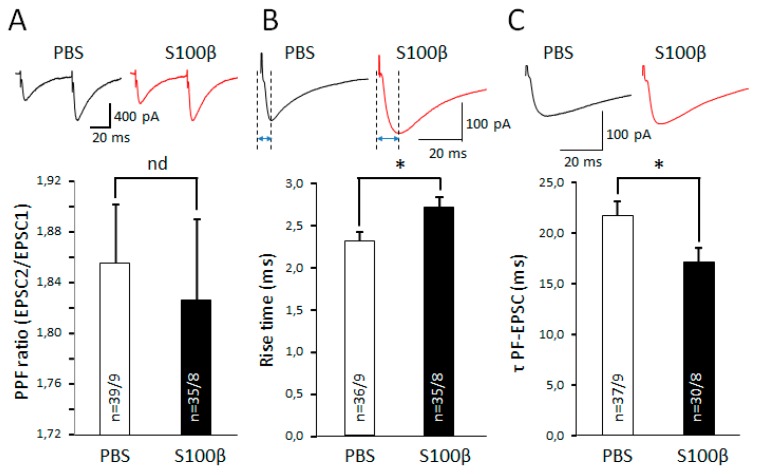
Alters PCs’ electrophysiological properties. (**A**) The summary graph shows the average PPF ratio (second amplitude/first amplitude EPSC) in PCs from PBS and S100β injected areas, no significant differences found. Below—representative traces of parallel fibre (PF)-EPSCs. (**B**) The summary graph shows the average rise time of PF-EPSCs in PCs from PBS and S100β injected areas. In comparison with PBS-injected areas, rise time was significantly longer in S100β injected areas. Representative traces of PF-EPSCs are shown above. (**C**) In comparison with PBS-injected areas, the decay time was significantly longer in S100β injected areas. The represented traces of PF-EPSCs are shown above. * *p* < 0.05.

**Figure 6 brainsci-09-00080-f006:**
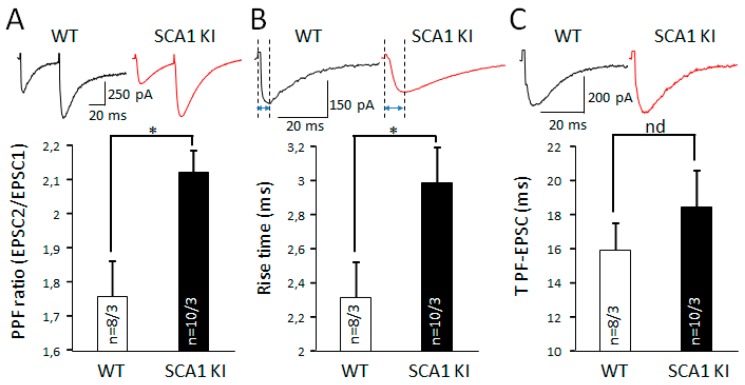
Properties of PCs are altered in three-week-old non-cell selective spinocerebellar ataxia type 1 (SCA1) model mice. (**A**) Average PPF ratio (second amplitude /first amplitude EPSC) in PCs from WT and SCA1 knock-in (KI) mice. In comparison with WT PCs, the PPF ratio significantly increased in SCA1 KI mice. Representative traces of PF-EPSCs are shown above. (**B**) The average rise time of PF-EPSCs in PCs from WT and SCA1 KI mice. In comparison with WT PCs, the rise time was significantly longer in SCA1 KI animals. Representative traces of PF-EPSCs are shown above. (**C**) Average decay time of PF-EPSCs in PCs from PBS and S100β injected areas. No significant differences. The represented traces of PF-EPSCs are shown above. * *p* < 0.05.

**Figure 7 brainsci-09-00080-f007:**
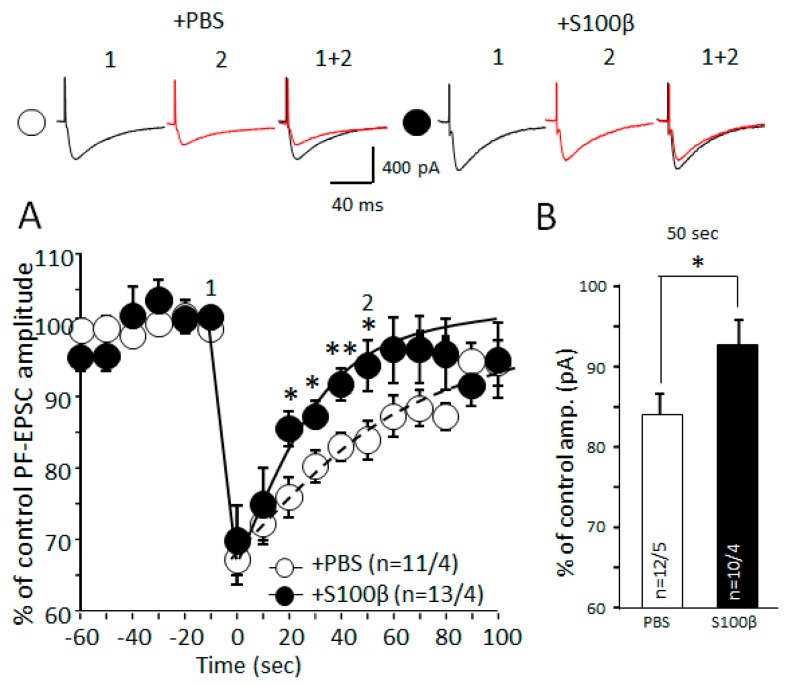
S100β disrupts the depolarization induced suppression of excitation (DSE) at PF–PC synapses. (**A**) Average time course diagram of PF-EPSC amplitudes before and after depolarization. The amplitudes of PF-EPSC were normalized to values before depolarization. The numbers (n) of tested PCs and animals (PCs/animals) are indicated in the graph. Dotted and black lines indicate the double waveform model fit for PBS and S100β injected groups, respectively. Representative PF-EPSC traces from PCs from PBS- and S100β-injected mice are shown above the diagram. Time points: before (1) and 50 s after (2) depolarization. (**B**) PF-EPSC amplitudes 50 s after depolarization. In comparison with PBS-injected areas, the PF-EPSC amplitude was significantly bigger in S100β injected areas. * *p* < 0.05.

**Figure 8 brainsci-09-00080-f008:**
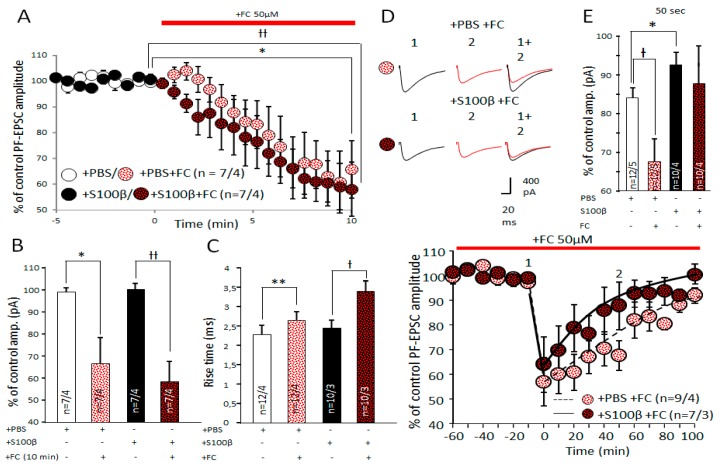
Downregulation of astrocytic function by fluorocitrate (FC) affects amplitude and kinetics of PF-EPSC in PF–PC synapses. (**A**) Time course of PF-EPSC amplitudes before and after 50 μM FC application. (**B**) Averaged PF-EPSC amplitudes 10 min after FC application. The effect of FC was approximately the same in S100β- and PBS-injected slices. (**C**) FC significantly changed PF-EPSC rise time in PBS-injected animals, and slightly increased it after administration of S100β. ** *p* < 0.01, Ɨ *p* < 0.05. (**D**) Representative PF-EPSC traces elicited in PCs from PBS and S100β-injected mice are shown above the diagram. Time points: before (1) and 50 s after (2) depolarization. (**E**) PF-EPSC amplitudes 50 s after depolarization. In comparison with PBS-injected areas, the PF-EPSC amplitude was significantly bigger in S100β injected areas. * *p* < 0.05 BG suppression by FC leads to significant DSE enchantment in PBS injected PCs. Ɨ < 0.05. (**F**) Average time course of PF-EPSCs before and after depolarization in slices treated with 50 μM FC. Dotted and black lines indicate the double waveform model fit for PBS-and S100β-injected groups, respectively.

**Figure 9 brainsci-09-00080-f009:**
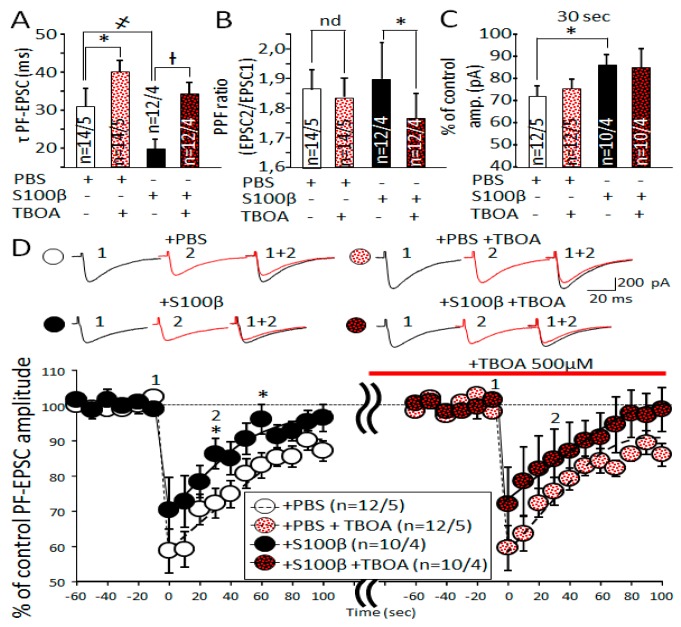
Inhibition of glutamate uptake by TBOA alters PF-EPSC kinetic, but does not affect endocannabinoid-mediated short term plasticity. (**A**,**B**) TBOA prolongs EPSC in PBS- and S100β-injected tissues (҂ *p* < 0.05). Note that the baseline tau was reduced after S100β application, but the effect of TBOA was comparable to the control. TBOA had a minimal effect on PPF ratio only in slices from S100β-injected animals. * *p* < 0.05. (**C**) Average diagram of PF-EPSC amplitudes before and 30 s after depolarization. Application of TBOA marked by red line. The amplitudes of PF-EPSC were normalized to values before depolarization. * *p* < 0.05. (**D**) Representative PF-EPSC traces elicited in PCs from PBS- and S100β-injected mice are shown above the diagram. Time points: before (1) and 30 s after (2) depolarization. Average time course of PF-EPSCs before and after depolarization in slices treated with TBOA. Dotted and black lines indicate the double waveform model fit for PBS- and S100β-injected groups, respectively.

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
