# Peer review of "Extracellular S100β Disrupts Bergman Glia Morphology and Synaptic Transmission in Cerebellar Purkinje Cells"

_brainsci, 2019, doi:10.3390/brainsci9040080_

Round 1
Reviewer 1 Report
Although this revised version is much improved from the original version, the immunohistochemical figures should be significantly improved. Because these photos can't express the morphology of the cells. Why are all images out of focus? If these images were obtained from the confocal microscope, that is unbelievable.
And I think Alexa-647 conjugated antibody is not suitable for this study. This fluorescence is indistinct, especially to observe at low magnification. I recommend the red fluorescence (e.g. Alexa-594) , because it usually used to make clear for merge image with Alexa-488 fluorescence image (double positive area express yellow fluorescence).
In conclusion, authors should be more concerned with making figures.
Author Response
Thank you very much for suggestion of confocal images. The most number of confocal images was made in z-stack mode. For this reason we succeeded to focus on BG processes labeled by anti-GFAP. Anti-S100b labeling did not give so clear image. It is due to S100b localization not only in processes but also in endfeet, soma etc. But anti-GFAP staining gives the clear picture. As you can see on Fig. 2A now the morphology of BG processes are clear to understand. In S100b injected animals they are thick and have intensive signal. Moreover we made Sholl analysis and on the Fig. 3A readers can easily understand the BG morphology.
Moreover “Fluoview” software for confocal Olympus microscope let us to change the colors of immunospecific signal. It was easy to change blue color to red. And we succeeded to get magenta yellow color (Fig. 1 and 2). We mentioned about it in methods.

Reviewer 2 Report
I do not have further comments.
Author Response
We are happy to know that our revision did not evoke any comments. Thank you very much. We made the minor remarks in manuscript. You can see it in final version.
This manuscript is a resubmission of an earlier submission. The following is a list of the peer review reports and author responses from that submission.
Round 1
Reviewer 1 Report
Introduction
The organization of this section should be improve.
1. Line 40-41, 45-46, 50-51, 60-61, 64-65: it is not necessary space in the beginning of the sentence and also making new line.
2. ATXN1(Line 39) and Ataxin 1(Line 55) are same or not? The author should be clear these (or this) gene(s).
3. It should be mentioned in detail about ‘various mouse models’ in line 54.
M & M
The author should describe in details. Overall, the explanation are insufficient.
1. The supplier of mice is not described.
2. The code number of all regents should be described.
3. Line 85: What is the injection method? The stereotaxic injection? If so, the detail position of bregma should be describe.
4. Line 91~: (in mM): 234 sucrose…. It should be replace to 234 mM sucrose. The same applies to the following (~Line 92. And also Line 95- 96……)
5. Line 136: The author should describe the detail conditions to get the confocal images. (Lens, zoom, pixel,…. et al.)
6. Line 138: What is the name and version of the ‘original software’? Which confocal did author use, Olympus or Zeiss?
7. Line 143: Where is the Fig 1F?
Results
Overall, the quality of the photographs is very low.
1. Fig 1A-B: These photos are almost meaningless. If the author want to show that ‘the area of s100b deposits spread much further than the actual site of injection… (Line 258-259)’, the readers never understand it from these photos. These photos should be more convincing.
2. Instead of the supplemental figure, the representative photos showing the result of Fig. 1C-E should be included in Fig.1.
3. Is there a difference of the cellular localization between exogenous and endogenous s100b expression? Both expression in cytoplasmic?
4. Fig 2: Are these figure legends correct? These legends don’t match with Figure 2.
5. Fig 2A is not clear.
Author Response
We want to thank all reviewers for critical comments. It helps us to improve significance of our data evidence. After additional experiments we increase the statistical samples. It helps us to estimate effect from S100b injection more accurately. Also we include data from 3 weeks old SCA1 knock-inn mice. It helps us to link S100b injection model with real SCA1 model. We also made the morphological analysis of BG using Sholl analysis. It helps us to see the alteration in S100b affected areas in single BG, what is impossible using routine IHC.
The organization of this section should be improve.
Introduction:
Q: 1. Line 40-41, 45-46, 50-51, 60-61, 64-65: it is not necessary space in the beginning of the sentence and also making new line.
A: Space was removed from mentioned lines.
Q: 2. ATXN1(Line 39) and Ataxin 1(Line 55) are same or not? The author should be clear these (or this) gene(s).
A: We pointed in the text that ATXN1 is gene and Ataxin1 is protein.
Q: 3. It should be mentioned in detail about ‘various mouse models’ in line 54.
A: We described 2 SCA1 mice models: transgenic B05 and SCA1 knock-inn. The data from SCA1 knock-inn mice were included in this work. All description about these models are with the references.
M & M
The author should describe in details. Overall, the explanation are insufficient.
Q: 1. The supplier of mice is not described.
A: We describe the supplier and the exact line of CD1 and SCA1 knock-inn mice
Q: 2. The code number of all regents should be described.
The code number of all blockers and S100b protein were included in the text.
Q: 3. Line 85: What is the injection method? The stereotaxic injection? If so, the detail position of bregma should be describe.
A: We stereotaxically injected PBS and S100b in cerebellar cortex. The coordinates according to the bregma was mentioned in the text.
Q: 4. Line 91~: (in mM): 234 sucrose…. It should be replace to 234 mM sucrose. The same applies to the following (~Line 92. And also Line 95- 96……)
A: We made this remarks.
Q: 5. Line 136: The author should describe the detail conditions to get the confocal images. (Lens, zoom, pixel,…. et al.)
A: We described the objective, zoom and pixels in the text. Also we put the detailed description of Z-stack microscopy for Sholl analysis.
Q: 6. Line 138: What is the name and version of the ‘original software’? Which confocal did author use, Olympus or Zeiss?
A: It was mentioned in line 126 that we used Olympus, Fluoview, FV10i (Japan). In line 139 we made slip of pen that mentioned about Zeiss microscope. This part was inprooved.
Q: 7. Line 143: Where is the Fig 1F?
A: All figures were reorganized. Remark is not actual now.
Results
Q: Overall, the quality of the photographs is very low.
A: All photos were converted in TIF format to improve its quality.
Q: 1. Fig 1A-B: These photos are almost meaningless. If the author want to show that ‘the area of s100b deposits spread much further than the actual site of injection… (Line 258-259)’, the readers never understand it from these photos. These photos should be more convincing.
A: We put the photos of whole cerebellar slice to show the injected areas with high fluorescent intensity after anti-S100b labeling and areas without injection. Now it should be clear that S100b deposits spread much futher that actual site of injection.
Q: 2. Instead of the supplemental figure, the representative photos showing the result of Fig. 1C-E should be included in Fig.1.
A: These data was replaced to the Fig. 2. The number of photos was enchanted dramatically and we decided to choose just one photo (Fig. 2B) that clearly describe the size, thickness of processes and number of Bergmann glia cell bodies. If this photograph is not clear, please point it.
Q: 3. Is there a difference of the cellular localization between exogenous and endogenous s100b expression? Both expression in cytoplasmic?
A: We suspect that after S100b injection this protein is located in extracellular space. But instead of earlier reports (Cvetanovich et al., 2015) we did not find S100b accumulation inside the Purkinje cells. They were negative for anti-S100b labeling.
Q: 4. Fig 2: Are these figure legends correct? These legends don’t match with Figure 2.
A: All figures were reorganized. Remark is not actual now.
Q: 5. Fig 2A is not clear.
A: We decided that the most unclear point in Fig. 2 was the thickness of molecular layer. We choose the photographs from PBS and S100b injected areas that has vividly different thickness of molecular layer (Fig. 4A). For better understanding we put a white dashed lines to mark molecular layer.

Reviewer 2 Report
This is an interesting report addressing the pathological role of exogenously injected S100b on Bergmann glia astrogliosis as Purkinje cell dysfunction. To the latter, the authors speculated that uptake of S100b and its Ca2+-chelating role could be causative to the neuronal dysfunction. Overall, the results generate pathological implication to the spinocerebellar ataxia type 1 (SCA1) where excessive excretion of S100b from reactive astrocytes and accumulation in Purkinje neurons has been demonstrated. The authors, however, need address the following points in revision.
1. Figure 1 legend: “antiS100b” and “antiGFAP” should read as “anti-S100b” and “anti-GFAP”, respectively. The bar graphs summarized the process thickness, number; it would be necessary to provide a set of representative images for both control and treated groups for illustration.
2. Fig 1B: do not see an S100b induced enhancement in GFAP expression, nor a quantitative comparison of GFAP staining between control and S100b treatment groups.
3. Ln 137 Fig. 1E does not match what is presented in the corresponding Fig 1. Also, in Ln 143: Fig. 1F is absent in Fig 1.
4. Ln 163: where is “Sup Fig 1A?
5. Ln 203-204: It is confusing here, you have vehicle PBS control, but what was the wildtype mice control? Did S100b injection speed up the decay, or PBS slowed down the decay? Please clarify. Do the results imply an elevated expression of glutamate transporters in Bergmann glia?
Author Response
We want to thank all reviewers for critical comments. It helps us to improve significance of our data evidence. After additional experiments we increase the statistical samples. It helps us to estimate effect from S100b injection more accurately. Also we include data from 3 weeks old SCA1 knock-inn mice. It helps us to link S100b injection model with real SCA1 model. We also made the morphological analysis of BG using Sholl analysis. It helps us to see the alteration in S100b affected areas in single BG, what is impossible using routine IHC.
Q: 1. Figure 1 legend: “antiS100b” and “antiGFAP” should read as “anti-S100b” and “anti-GFAP”, respectively. The bar graphs summarized the process thickness, number; it would be necessary to provide a set of representative images for both control and treated groups for illustration.
A: “antiS100b” and “antiGFAP” replaced to “anti-S100b” and “anti-GFAP”. These data was replaced to the Fig. 2. The number of photos was enchanted dramatically and we decided to choose just one photo (Fig. 2B) that clearly describe the size, thickness of processes and number of Bergmann glia cell bodies. If this photograph is not clear, please point it.
Q: 2. Fig 1B: do not see an S100b induced enhancement in GFAP expression, nor a quantitative comparison of GFAP staining between control and S100b treatment groups.
A: We put the photos of whole cerebellar slice to show the injected areas with high fluorescent intensity after anti-S100b labeling and areas without injection (Fig. 1). Before confocal microscopy we set standard properties of laser intensity and sensitivity (see methods) to the certain level. So, the intensity of fluorescence should reflect the actual enhancement in S100b and GFAP expression.
Q: 3. Ln 137 Fig. 1E does not match what is presented in the corresponding Fig 1. Also, in Ln 143: Fig. 1F is absent in Fig 1.
A: All figures were reorganized. Remark is not actual now. The information about that please see in Fig. 2.
Q: 4. Ln 163: where is “Sup Fig 1A?
A: We attached all Sup. Figures with revised manuscript.
Q: 5. Ln 203-204: It is confusing here, you have vehicle PBS control, but what was the wildtype mice control? Did S100b injection speed up the decay, or PBS slowed down the decay? Please clarify. Do the results imply an elevated expression of glutamate transporters in Bergmann glia?
A: We made additional experiments with non-injected mice and did not see any changes of PF-EPSC decay time in naïve Purkinje cells in comparison to PBS injected Purkinje cells (see Sup. Fig. 3). Also we made additional experiments with selective blockers of EAATs (DL-TBOA). We confirmed that S100b enchase glutamate reuptake through these transporters because TBOA sweep out difference in decay time between PBS and S100b injected areas (Fig. 9B). Our findings are confirming the earlier findings (Tramontina et al., 2006). The description please sees in the text. We cannot say that S100b leads to increasing of EAATs expression. It is better to prove this hypothesis by RT-PCR, than by routine IHC that we used here. We are planning to do these experiments in future.

Reviewer 3 Report
The paper by Belozor et al. investigates the effects of extracellular S100b protein on morphology and function of glial and neuronal cells of cerebellum. The S100b protein is universally acknowledged as specific astroglial marker but its roles in health and disease are not fully understood. So, the data reported in the manuscript could be interesting for wide audience of neuroscientists working in the area of glia-neuron interactions. The work has been performed at good technical quality and, in general, is suitable for publication. Yet, there are few issues addressing of which may significantly increase the quality and impact of the article.
Specific points
1. Abstract, line 22 – remove “endemic in some part of Siberia” – this may be misleading for readers unfamiliar with epidemiology and pathophysiology of Spinocerebellar ataxia and undermine the importance of the paper for wider audience. Actually, the SCA1 accounts for 6-27% of all dominant ataxias and has rather high prevalence in Poland and South Africa. Also, one cannot exclude involvement of S100b in pathophysiology of other types of spinocerebellar ataxia. Anyway, such details should be put in the discussion rather than abstract.
2. In the section 3.1 Authors show some data alterations in the Bergmann glia organisation and suggest ( in the Introduction and Discussion) that these changes are important since they can affect glial uptake of extracellular glutamate. This part of the paper could be strengthened by providing a deeper insight into BG cells morphology, in particular, changes in arborisation. The cell arborisation can be analysed by basic circular Sholl analysis, e.g. implemented in the ImageJ/FIJI software (see https://imagej.net/Sholl_Analysis). The output of such analysis could provide important parameters which would be more relevant to “glia coverage” of synapses than linear density of processes currently shown in the Fig.1. Extending section 3.1 by showing examples of individual BG cells and results of Sholl analysis would significantly increase the quality and importance of results. This analysis would not even require additional experiments, just revisiting available immunostaining preparations.
3. It is pity that Authors did not accompany their results on changes in the BG cells morphology by any functional data. Such data might include basic electrophysiological assessment of hallmark glial cell functions: 1) density of potassium currents 2) EAAT-mediated currents evoked in the glial cells by application of glutamate. The latter might be of a great importance for the overall message of the paper since Authors view alterations in the glial uptake of glutamate as mechanism underlying changes in synaptic transmission.
The research team seems to be well equipped and capable for such kind of recordings.
4. Authors show interesting data on S100b-induced changes in the cannabinoid-dependent short-term plasticity (DSE) in Purkinje cells but interpret their results based on rather out-dated “presynaptic model” of DSE (lines 220-221). Such interpretation presumes only remote involvement of glial cells and therefor undermines a putative impact of the paper on research field of glia-neuron interactions. However, endocannabinoids released from postsynaptic cells after VG Calcium channels activations could also directly activated CB1 receptors in glial cells (Rasooli-Nejad, Philos Trans R Soc Lond B 2014, Metna-Laurent Glia 2016), leading to Ca-elevation which can affect many basic astrocytic functions. Such interpretation goes well in line with Authors’ idea of alterations in glial uptake of glutamate. Also, lack of significant changes in the probability of presynaptic glutamate release (Fig.3A) argues against involvement of pure “neuronal” mechanism of DSE but changes in the EPSC decay time (Fig.3B) and DSE time course support the changes in glutamate uptake.
One might suggest some additional experiments here to substantiate the involvement of glial mechanisms to DSE: 1) testing whether inhibition of glia cell functions with glia metabolic poisons fluorocitrate/fluoroacetate occludes the effect of S100b on DSE; 2) testing the effects of EAAT1 (which are expressed mainly in glia) antagonists on EPSC and DSE.
Performing of these exepriments is advisory but extending of the Discussion by addressing the putative involvement of glial CB1 signalling is essential.
5. In the Discussion, line 309, Authors mention “own unpublished data” on effect of Atxn1 mutation on astrocytes. This paper seems to be a good place to publish these data. Could Authors provide some data here, even as Supplementary? This would strengthen the message and impact of the paper?
If not, the mentioning of the data should be removed from the refences, i.e. mention only literature data from the ref. [42]
6. The presented values of PC capacitance (about 400 pF) seems to be a way too high to reflect “predominately size of the soma”. The soma diameter of 20-30 mkm would have rendered the capacitance of 30-50 pF, typically measured in the enzymatically isolated Purkinje cells devoid of most of dendritic tree. This discrepancy brings up the question: how was cell capacitance measured (this should be pointed out in Methods) – by compensating the slow capacitance via amplifier (and using red-outs) or by decay time of the cell recharge transient ? Second approach is more accurate and informative, it can also pick-up an involvement of dendritic capacitance which would manifest in the appearance of secondary, much slower, exponential function in the decay profile (see Major, G., J. Evans, and J. J. B. Jack. 1993. Biophys. J. 65:450– 468 for some theory). Since the data on cell capacitance were on the brink of statistical significance (Fig.2B), more accurate analysis would be beneficial because it might reveal the S100b-induced changes in the dendritic arborisation. Furthermore, it could also be accompanied by the Sholl analysis of PC arborisation.
7. The paper would certainly benefit from thorough check and editing of English.
Here several examples of Language/Style/Grammar errors needing correction:
line 135, Methods, – words “Analysis of Bergmann glia morphology” should be italized and the following paragraph (ln.135-144) should have made a sub-section (i.e. 2.4);
lines 176-179 (Results 3.2 ) Change for “The capacitance of PCs in the S100b mice was…. and 461.1±24.8 pF in the PBS injected group. However, the difference between two groups was not significant accordingly Mann-Whitney test”;
lines 184-185 (Fig.2 legend) Change for “The capacitance of PCs was measured using voltage-clamp, no difference between PBS and S100b injected area of cerebellar cortex was found”;
lines 196-197, change for “There was no statistically significant difference in the amplitude and rise time of EPSCs recorded in the PCs of mice injected with S100p and PBS”
line 216, change “suspected” for …” expected”;
lines 252-253, the sentence starting from “In SCA1 mouse, S100b contains cytoplasmic vacuoles” sounds confusing. Should it read “…S100b-containing vacuoles”, perhaps?
Lines 263-264, change for “To avoid false positive results when measuring the BG and PC morphology, we used the lobes 6 and 7 in all experiments”.
Lines 277-278, change “For this reason” for “Hence”;
Lines 281-282, change for “Accordingly our observations, the S100b did not affect the presynaptic glutamate release since it did not significantly change the PPF ratio”;
Line 289, change for “We assessed the effect of S100b on short-term synaptic plasticity by alterations in the DSE”.
Author Response
We want to thank all reviewers for critical comments. It helps us to improve significance of our data evidence. After additional experiments we increase the statistical samples. It helps us to estimate effect from S100b injection more accurately. Also we include data from 3 weeks old SCA1 knock-inn mice. It helps us to link S100b injection model with real SCA1 model. We also made the morphological analysis of BG using Sholl analysis. It helps us to see the alteration in S100b affected areas in single BG, what is impossible using routine IHC.
Q: 1. Abstract, line 22 – remove “endemic in some part of Siberia” – this may be misleading for readers unfamiliar with epidemiology and pathophysiology of Spinocerebellar ataxia and undermine the importance of the paper for wider audience. Actually, the SCA1 accounts for 6-27% of all dominant ataxias and has rather high prevalence in Poland and South Africa. Also, one cannot exclude involvement of S100b in pathophysiology of other types of spinocerebellar ataxia. Anyway, such details should be put in the discussion rather than abstract.
A: We are agreeing that SCA1 is Worldwide pathology. We removed the phrase about SCA1 endemism in Siberia.
Q: 2. In the section 3.1 Authors show some data alterations in the Bergmann glia organisation and suggest ( in the Introduction and Discussion) that these changes are important since they can affect glial uptake of extracellular glutamate. This part of the paper could be strengthened by providing a deeper insight into BG cells morphology, in particular, changes in arborisation. The cell arborisation can be analysed by basic circular Sholl analysis, e.g. implemented in the ImageJ/FIJI software (see https://imagej.net/Sholl_Analysis). The output of such analysis could provide important parameters which would be more relevant to “glia coverage” of synapses than linear density of processes currently shown in the Fig.1. Extending section 3.1 by showing examples of individual BG cells and results of Sholl analysis would significantly increase the quality and importance of results. This analysis would not even require additional experiments, just revisiting available immunostaining preparations.
A: We want to thank the reviewer for essential note that increase the provement of our hypothesis. We made the Sholl analysis and found the alteration of BG morphology (Fig. 3 and explanation in the text).
Q: 3. It is pity that Authors did not accompany their results on changes in the BG cells morphology by any functional data. Such data might include basic electrophysiological assessment of hallmark glial cell functions: 1) density of potassium currents 2) EAAT-mediated currents evoked in the glial cells by application of glutamate. The latter might be of a great importance for the overall message of the paper since Authors view alterations in the glial uptake of glutamate as mechanism underlying changes in synaptic transmission.
The research team seems to be well equipped and capable for such kind of recordings.
A: Thank you very much for this note. We have all necessary equipment in our laboratory, but we are not experts in astrocytic patch. We are planning to make these essential experiments in future. Solid data of true evident results of BG electrophysiological properties takes time – several months.
Q: 4. Authors show interesting data on S100b-induced changes in the cannabinoid-dependent short-term plasticity (DSE) in Purkinje cells but interpret their results based on rather out-dated “presynaptic model” of DSE (lines 220-221). Such interpretation presumes only remote involvement of glial cells and therefor undermines a putative impact of the paper on research field of glia-neuron interactions. However, endocannabinoids released from postsynaptic cells after VG Calcium channels activations could also directly activated CB1 receptors in glial cells (Rasooli-Nejad, Philos Trans R Soc Lond B 2014, Metna-Laurent Glia 2016), leading to Ca-elevation which can affect many basic astrocytic functions. Such interpretation goes well in line with Authors’ idea of alterations in glial uptake of glutamate. Also, lack of significant changes in the probability of presynaptic glutamate release (Fig.3A) argues against involvement of pure “neuronal” mechanism of DSE but changes in the EPSC decay time (Fig.3B) and DSE time course support the changes in glutamate uptake.
One might suggest some additional experiments here to substantiate the involvement of glial mechanisms to DSE: 1) testing whether inhibition of glia cell functions with glia metabolic poisons fluorocitrate/fluoroacetate occludes the effect of S100b on DSE; 2) testing the effects of EAAT1 (which are expressed mainly in glia) antagonists on EPSC and DSE.
Performing of these exepriments is advisory but extending of the Discussion by addressing the putative involvement of glial CB1 signalling is essential.
A: Thank you very much for advice! We made additional experiments with fluorocitrate and TBOA. Data from these experiments help us to show the influence of BG to DSE. Totally we used nearly 10-12 mice and 40 cells in each group. Increasing of statistical sampling leads to helps us to estimate the influence of S100b to PCs capacitance, PF-EPSC rise and decay time.
Q: 5. In the Discussion, line 309, Authors mention “own unpublished data” on effect of Atxn1 mutation on astrocytes. This paper seems to be a good place to publish these data. Could Authors provide some data here, even as Supplementary? This would strengthen the message and impact of the paper?
If not, the mentioning of the data should be removed from the refences, i.e. mention only literature data from the ref. [42]
A: Unfortunatelly we can not publish our own data about DSE impairment in 12 weeks old SCA1 knock-inn mice. We are pnanning to publish it in separate paper with collaboration of our Japanese colleges. But We after analyzing data from 3 weeks old SCA1 knock-inn mice we found altered PF-EPSC kinetics which could be similar to alterations in PCs affected exogenous S100b. We include these data in this paper. This data helps to link pathology in S100b injected mice with those of SCA1 model mice. It also enchance the discussion and made the solid inmplications.
Q: 6. The presented values of PC capacitance (about 400 pF) seems to be a way too high to reflect “predominately size of the soma”. The soma diameter of 20-30 mkm would have rendered the capacitance of 30-50 pF, typically measured in the enzymatically isolated Purkinje cells devoid of most of dendritic tree. This discrepancy brings up the question: how was cell capacitance measured (this should be pointed out in Methods) – by compensating the slow capacitance via amplifier (and using red-outs) or by decay time of the cell recharge transient ? Second approach is more accurate and informative, it can also pick-up an involvement of dendritic capacitance which would manifest in the appearance of secondary, much slower, exponential function in the decay profile (see Major, G., J. Evans, and J. J. B. Jack. 1993. Biophys. J. 65:450– 468 for some theory). Since the data on cell capacitance were on the brink of statistical significance (Fig.2B), more accurate analysis would be beneficial because it might reveal the S100b-induced changes in the dendritic arborisation. Furthermore, it could also be accompanied by the Sholl analysis of PC arborisation.
A: We carefully analyzed our passive electrophysiological properties of PCs according to method, described by Major et al. We modernized the calculation of soma and dendritic capacitance (see the methods) and find vivid argument of PCs alteration after S100b injection (Fig. 4C).
Q: 7. The paper would certainly benefit from thorough check and editing of English.
Here several examples of Language/Style/Grammar errors needing correction:
line 135, Methods, – words “Analysis of Bergmann glia morphology” should be italized and the following paragraph (ln.135-144) should have made a sub-section (i.e. 2.4);
A: We made the separate subsection for Sholl analysis.
Q: lines 176-179 (Results 3.2 ) Change for “The capacitance of PCs in the S100b mice was…. and 461.1±24.8 pF in the PBS injected group. However, the difference between two groups was not significant accordingly Mann-Whitney test”;
A: This phrase was modified.
Q: lines 184-185 (Fig.2 legend) Change for “The capacitance of PCs was measured using voltage-clamp, no difference between PBS and S100b injected area of cerebellar cortex was found”;
A: Data was changed. This phrase is not actual now.
Q: lines 196-197, change for “There was no statistically significant difference in the amplitude and rise time of EPSCs recorded in the PCs of mice injected with S100p and PBS”
A: This phrase was modified.
Q: line 216, change “suspected” for …” expected”;
A: This phrase was modified.
Q: lines 252-253, the sentence starting from “In SCA1 mouse, S100b contains cytoplasmic vacuoles” sounds confusing. Should it read “…S100b-containing vacuoles”, perhaps?
A: Thank you for correction. We changed to S100b-containing vacuoles.
Q: Lines 263-264, change for “To avoid false positive results when measuring the BG and PC morphology, we used the lobes 6 and 7 in all experiments”.
A: This phrase was modified.
Q: Lines 277-278, change “For this reason” for “Hence”;
A: This word was changed.
Q: Lines 281-282, change for “Accordingly our observations, the S100b did not affect the presynaptic glutamate release since it did not significantly change the PPF ratio”;
A: This phrase was modified.
Q: Line 289, change for “We assessed the effect of S100b on short-term synaptic plasticity by alterations in the DSE”.
A: This phrase was modified.